# Health Literacy among Japanese College Students: Association with Healthy Lifestyle and Subjective Health Status

**DOI:** 10.3390/healthcare11050704

**Published:** 2023-02-27

**Authors:** Hisayo Yokoyama, Daiki Imai, Yuta Suzuki, Akira Ogita, Hitoshi Watanabe, Haruka Kawabata, Takaaki Miyake, Izumi Yoshii, Shinji Tsubouchi, Yoshimasa Matsuura, Kazunobu Okazaki

**Affiliations:** 1Research Center for Urban Health and Sports, Osaka Metropolitan University, 3-3-138 Sugimoto, Sumiyoshi-ku, Osaka-shi, Osaka 558-8585, Japan; 2Department of Environmental Physiology for Exercise, Osaka Metropolitan University Graduate School of Medicine, 3-3-138 Sugimoto, Sumiyoshi-ku, Osaka-shi, Osaka 558-8585, Japan

**Keywords:** health literacy, web-based survey, collegiate health, health promotion, disease prevention, health education, primary healthcare

## Abstract

The improvement of health literacy (HL) is a critical issue for college students who are in the transitional period to adulthood and are establishing their subsequent lifestyles. The present study aimed to evaluate the current state of HL among college students and to explore the factors that influence HL. Moreover, it investigated the relationship between HL and health conditions. For this study, the researchers conducted an online survey of college students. The questionnaire consisted of the Japanese version of the 47-item European Health Literacy Survey Questionnaire (HLS-EU-Q47), which is a self-assessment tool for HL that covers the major health issues of college students and health-related quality of life. The study analyzed 1049 valid responses. Based on the HLS-EU-Q47 total score, 85% of the participants exhibited *problematic* or *unsatisfactory* HL levels. Participants who reported high levels of healthy lifestyles obtained high HL scores. High levels of HL were associated with high levels of subjective health. Results from quantitative text analysis suggested that specific mindsets were correlated with high levels of competency in appraising health information among male students. In the future, educational intervention programs for college students need to be established to improve HL levels.

## 1. Introduction

As the world faced unprecedented challenges during the COVID-19 pandemic, it also witnessed social unrest through social media. The displayed behaviors included food hoarding at supermarkets, prejudice and discrimination against medical staff and their family members, and excessive reactions to reports of the side effects of vaccines. Such situations are a reminder of the importance of obtaining accurate health information and making informed decisions before taking appropriate actions. However, such decision-making is not necessarily easy in the current situation, in which a tremendous amount of information is being disseminated through the Internet and social networking services by people who lack expertise.

According to the Oxford English Dictionary (https://www.oed.com/ accessed on 30 November 2022), literacy originally referred only to the ability to read and write. As the disease structure changed over time, even in developing countries (i.e., from communicable to noncommunicable diseases, such as diabetes), literacy clearly became the cornerstone of disease prevention and health management. Therefore, scholarly attention focused on literacy regarding health or health literacy (HL). In 2012, Sørensen et al. [1] defined HL as follows:


…people’s knowledge, motivation, and competencies to access, understand, appraise, and apply health information in order to make judgments and take decisions in everyday life concerning healthcare, disease prevention, and health promotion to maintain or improve quality of life during the life course.


Now, the concept of HL has become widely known. In other words, HL is a term that includes not only decision-making but also problem-solving skills in healthcare, disease prevention, and health promotion.

HL is required when addressing not only the COVID-19 pandemic but also various daily situations, to enable individuals to manage their health on their own initiative. The results of systematic reviews suggest the various effects of HL on health outcomes. Particularly, the lower the level of HL, the higher the rates of hospitalization and emergency outpatient use and the fewer the opportunities for cancer screening and vaccination [2]. Older adults tend to display poor overall health status and high mortality rates [2]. Low levels of HL also result in additional medical costs [3]. Conversely, high levels of HL promote healthy lifestyle habits, such as the increased consumption of vegetables and fruits and less smoking [4].

The Japanese Ministry of Health, Labor, and Welfare is working to maintain the world’s top level of health, in the face of a declining birthrate and an aging population, and to prevent future generations from being burdened by mounting medical costs. In 2015, it envisioned ideal healthcare policies for Japan and proposed “Health Care 2035” [5]. In this proposal, the Ministry mentions that promoting improvement in HL across the nation is important, such that citizens can proactively select the medical services that are suitable for them in cooperation with medical professionals [6]. However, at present, the government or schools lack such opportunities for acquiring HL through education or such efforts from medical professionals and insurers.

College students in the transitional period to adulthood are becoming independent from their parents and are establishing their subsequent lifestyles. At the same time, students are responsible for their education, face various conflicts, and experience health problems specific to the learning process as they establish a wider range of social and human relationships than ever before in their university life [7,8,9]. Thus, improving the HL of college students is expected not only to contribute to their health but also to promote the health of society through their future careers. In 2021, 82.9% of Japanese individuals used the Internet [10]; therefore, promoting the provision of health information via the Internet is necessary. However, because the vast range of health information available on the Internet is not necessarily reliable, college students need to acquire the skills to judge the health information obtained from the Internet for themselves. Therefore, elucidating the current situation of HL among college students and the resources for such health information would be beneficial for the construction of new schemes for health education that center on HL. However, few studies have reported on the HL of college students.

To address this concern, the present study intends to evaluate the current state of HL among Japanese university students. It also examines the factors that influence HL and explores the relationship between HL and health conditions.

## 2. Method

### 2.1. Participants

The study recruited students between the ages of 18 and 29 years. They were currently enrolled at Osaka City University (https://www.osaka-cu.ac.jp/en [11] accessed on 16 November 2022) and Osaka Prefecture University (https://www.osakafu-u.ac.jp/en/ [12] accessed on 16 November 2022); both universities were merged into Osaka Metropolitan University in April 2022 (https://www.omu.ac.jp/en/ accessed on 16 November 2022) [13]. The exclusion criterion was difficulty in participating in the online survey using smartphones. The Ethical Committee of the Graduate School of Medicine of Osaka City University approved the research protocol (approval number: 2020-206). Students who met the abovementioned eligibility criterion were invited to participate through a learning management system. After reading the document that explained the objective of the study on a website, consent was considered to be given if the participants had ticked on the check column.

### 2.2. Procedure

This study is cross-sectional in nature and was conducted using a web survey site designed for its objective in December 2020 and October 2021. The questionnaire consisted of six sections, namely, demographic characteristics, sources of information on health, the content of health information from the Internet, self-perceived HL, common health problems, and health-related quality of life. In addition, we set an open-ended question in which the participants can freely describe the aspects they consider when obtaining or using health information (including that on the Internet). They were allowed to skip any question. The time required to answer the questionnaire was ~30 min. We entrusted the design of the web questionnaire and the collection of the responses to Data Select Co., Ltd. (Toyoake City, Aichi, Japan).

### 2.3. Self-Perceived HL

The study assessed HL using the Japanese version of the 47-item European Health Literacy Survey Questionnaire (HLS-EU-Q47) [14]. It is a self-report rating scale of HL developed by the European Health Literacy Survey, which can comprehensively evaluate not only HL at the individual level but also social policies and the local environment [15]. Nakayama et al. [14] confirmed the reliability and validity of the Japanese version of the HLS-EU-Q47. It assesses four competencies in processing health information and decision-making in health issues, namely, accessing, understanding, appraising, and applying. These competencies are assessed across three domains, namely, healthcare (medical care), disease prevention, and health promotion. The 47 items are rated using a five-point Likert-type scale (*very difficult* = 1, *fairly difficult* = 2, *fairly easy* = 3, *very easy* = 4, and *don’t know/not applicable*). The study coded the response *don’t know/not applicable* as a missing value, and results with missing values in even one of the 47 items were excluded from the analysis. Based on the method of the developer [15], the mean scores of each competency in the three domains were standardized to a maximum of 5 points using the following formula: (mean − 1) × 5/3. The total score of the 47 items was also standardized to a maximum of 50 points using the formula: (mean − 1) × 50/3. Furthermore, the standardized total scores were used to categorize HL levels, as follows: *inadequate* (0–25 points), *problematic* (>25, ≤33 points), *sufficient* (>33, ≤42 points), and *excellent* (>42, ≤50 points).

### 2.4. Health Challenges Common to College Students

Aiming to investigate the relationship between lifestyle habits that affect health conditions and HL, we investigated the common health challenges faced by college students. The American College Health Association (ACHA) (https://www.acha.org/ [16] accessed on 5 June 2020) regularly conducts a National College Health Assessment (https://www.acha.org/ncha [17] accessed on 5 June 2020), which is a national survey on the health of college students from various institutions across the United States. The ACHA identifies critical health issues for college students based on the results of the survey [18]. With reference to the critical health issues for college students suggested by the ACHA, we formulated seven questions on general health, exercise habits, smoking, sleep, vaccination, diet, and friendship. Items were rated using a five-point Likert-type scale (from *strongly disagree* to *strongly agree*).

### 2.5. Health-Related Quality of Life

Aiming to investigate the relationship between HL and subjective health awareness or comprehensive health status, we investigated the health-related quality of life of the participants. The Japanese version of the short form, (SF)-36, was used to evaluate comprehensive health status, which is one of the most widely and internationally used comprehensive health scales. Fukuhara et al. [19] confirmed the reliability and validity of the Japanese version. It consists of 36 items for evaluating eight dimensions of health (subdomains; see Table 1), which are rated using a Likert-type scale. Based on raw scores, the study calculated norm-based scores (from 0 = worst to 100 = best) using a web-based scoring system provided by iHope International Inc., in which the mean and standard deviation of the Japanese national standard value in 2007 were set as 50 and 10, respectively [20].

### 2.6. Quantitative Text Analysis

Among the open-ended responses, text data were quantified using the text mining method via language analysis software, Text Mining Studio version 6.4 (NTT Data Mathematical Systems, Tokyo, Japan). First, the researchers performed a morphological analysis of the text data by dividing sentences into the smallest grammatically meaningful components. A word relationship network was then created to visualize the characteristic terms that frequently appear in relation to gender. In this figure, arrows connected the co-occurring terms (frequently used together in sentences and exhibiting similar appearance patterns), which were then divided into several clusters. The stronger the degree of co-occurrence, the thicker the arrow; the larger the circle, the more frequent the appearance of a word [21]. Furthermore, the co-occurrence network schema was used to categorize the text data. We referred back to the original sentences containing terms that comprised a cluster to elucidate issues when obtaining and analyzing health information by gender [22,23,24].

### 2.7. Statistical Analysis

The researchers examined all variables for normal distribution using the Kolmogorov–Smirnov test. They performed comparisons between the two groups using an unpaired *t*-test and the Mann–Whitney U-test for normal and non-normal distribution, respectively. In addition, the study used Spearman’s correlation analysis to explore the correlation between variables, including categorical ones, and the χ^2^ test to compare the proportions of each item between the two groups. The Mantel–Haenszel test was used to identify trends between demographic characteristics and the levels of HL. Data were analyzed using SPSS version 27.0 (IBM Corporation, Armonk, NY, United States). A *p*-value of < 5% was considered statistically significant.

## 3. Result

### 3.1. Characteristics and Acquisition Status of Health Information

A total of 1322 students (98.1%), out of the 1347 students invited to participate in the web survey through the classes, agreed to participate. Responses were excluded if the age of the participants was unknown, or if the results of the HLS-EU-Q47 contained incomplete or missing values. After screening the responses, responses from 1049 participants (male: 623, female: 413; preferred not to respond: 13) were used for analysis. First-year undergraduate students accounted for 63.5% of the participants, and the average age of the entire sample was 19.8 ± 1.9 years (standard deviation (SD); see Table 2). The participants were distributed across eight faculties, including humanities and social sciences, science, and medical faculties. The gender distribution also represented the Japanese college student population.

Figure 1 presents the results of the received responses to the question, “Which information source do you use when obtaining health information? (multiple choices allowed).” The majority used the Internet, including social media, as sources of health information, followed by friends/family members and TV/radio. Specific responses to “others” included university classes, academic papers, and ASMILE (a health app administered by the Osaka prefecture; https://www.asmile.pref.osaka.jp/ accessed on 14 January 2021), among others. Figure 2 depicts the results of the responses to the question, “What kind of health information do you obtain from the Internet? (multiple choices allowed).” The majority pointed to information on health management and disease prevention, followed by exercise and fitness, and therapeutic approaches for a disease.

### 3.2. Current State of HL

Table 3 displays the scores for each competency in the three domains of HLS-EU-Q47. Among the four competencies, *appraising* in all domains and *applying* in the field of health promotion resulted in lower scores, compared with those of other items. The average total score for HLS-EU-Q47 was 26.3 ± 7.7 points (out of 50 points); ~85% of the participants were categorized under an *inadequate* or *problematic* level of HL. Although no difference was noted in the total scores for HLS-EU-Q47 by gender (26.5 ± 8.3 points for male students, 26.0 ± 6.7 points for female students, *p* = 0.301), the distribution ratio of the levels of HL differed between men and women, in which the rates for *inadequate* and *problematic* were higher for female than for male students (*p* = 0.013). In addition, the distribution ratio differed according to their year of study, with *inadequate* being particularly high among first-year students (*p*-value for the trend < 0.001, Figure 3). Alternatively, the study observed no significant relationships in HL levels with the type of residence (living with family members, living alone, or living in a dormitory), annual household income, the presence or absence of current illnesses under treatment, or exercise habits.

### 3.3. Relationships between Common Health Problems among College Students and HL

We examined the relationship between responses to questions about the major health issues of college students, as proposed by the ACHA, and the total scores for the HLS-EU-Q47. Participants who responded that they subjectively had a higher general health status showed higher total scores for HLS-EU-Q47 than those who did not (ρ = 0.128, *p* < 0.001). Similarly, participants who responded that they had acquired or practiced healthy lifestyle habits in terms of sleep (ρ = 0.078, *p* = 0.011), eating (ρ = 0.070, *p* = 0.024), and maintaining friendships (ρ = 0.064, *p* = 0.037) had higher total scores for HLS-EU-Q47 than those who did not. Surprisingly, the participants who answered that they had received the necessary vaccinations obtained lower total scores for HLS-EU-Q47 than those who did not (ρ = −0.097, *p* = 0.002). Neither the response to the question about regular physical activity nor about smoking habits were correlated to the total scores for HLS-EU-Q47 (see Appendix A).

### 3.4. Association between HL Level and Health-Related QoL

We investigated the relationship between the levels of HL and calculated norm-based scores for each subdomain of the SF-36. The results indicated that high levels of HL were associated with high norm-based scores for physical functioning (ρ = 0.068, *p* = 0.027), role—physical (ρ = 0.121, *p* < 0.001), bodily pain (ρ = 0.091, *p* = 0.003), general health perceptions (ρ = 0.112, *p* < 0.001), vitality (ρ = 0.104, *p* < 0.001), role—emotional (ρ = 0.091, *p* = 0.003), and mental health (ρ = 0.078, *p* = 0.011). However, HL levels were not associated with a norm-based score for social functioning (See Appendix A).

### 3.5. Quantitative Text Analysis of Open-Ended Responses

A total of 368 participants provided responses to the open-ended question, “What aspects do you carefully consider on a daily basis or find difficult when obtaining and applying health-related information (including that from the Internet)?” Table 4 presents basic data regarding the sentence composition of text data. To identify the terms that appeared in the responses, we conducted a frequency analysis to create a list of the top 20 most frequently used terms (Table 5). Specific items related to the content of information to be obtained, such as “disease” and “symptoms,” and those related to personal mindset when accessing and evaluating information, such as “multiple (websites),” “be careful,” and “confirmation,” frequently appeared. Terms that expressed opinions about health information, such as “(the amount of information is too) much” and “(it is difficult to judge) credibility,” were also included.

Figure 4 depicts the co-occurrence network schema. Since a difference in the distribution ratio of the levels of HL between males and females was found, we mapped the relationship between gender and the terms used to explore the characteristics of the responses by gender. We grouped terms with similar appearance patterns into several clusters. When referring to the original sentences in the responses on the basis of these clusters, we found that an extremely common opinion for the male and female students references the difficulty found in distinguishing reliable information in the vast amount of medical and health-related information on the Internet. As a characteristic of their responses, the male students provided many descriptions of specific behaviors or mindsets for improving HL, such as “comparing multiple pieces of information.” Similarly, many male participants responded that they “always look at information with a critical eye” and “always check the source of the information.” Conversely, many female students responded with “I’m not sure that information is believable, except in the case where the credibility of it is clearly questionable” and “I can’t judge whether information is applicable to me” (Table 6).

## 4. Discussion

The majority of the participants responded that they used the Internet, including social media, as the main source of health information. Female and lower-grade students were more likely to exhibit low levels of HL. In addition, these levels were low in comparison with those of college students in other countries. In particular, the scores for the HLS-EU-Q47 were low in the competencies *appraising health information* in all fields and *applying health information* in the field of health promotion. In contrast, high levels of HL were associated with positive lifestyle habits for health maintenance and positive subjective health status.

HL is the ability to make decisions about health maintenance and management, based on health information [1]. Researchers demonstrate that insufficient HL exerts a negative impact on health outcomes. People with inadequate HL are less likely to receive preventive services, such as health checkups [2], lack knowledge about diseases and treatments [25], experience difficulty in communicating their symptoms and concerns to medical staff [26], exhibit high rates of hospitalization and mortality [2], and are consequently obliged to spend extra money on medical costs [3]. In the current study, high levels of HL were associated with positive lifestyle habits for health maintenance and high subjective health status. This result is consistent with those of the abovementioned studies. In particular, for college students who are in the process of becoming socially and economically independent from their parents, improving HL is critical for acquiring and maintaining a healthy lifestyle for the rest of their lives.

The levels of HL of the participants were low, and the distribution ratio of the levels of HL was similar to those from a previous survey by Nakayama et al., who reported that the levels of HL among the Japanese were lower than those of people in European countries (Figure 5A) [14]. Likewise, this pattern was observed when comparing the results with those of surveys that evaluated the HL levels of college students in other countries, using the HLS-EU-Q47 (Figure 5B) [27,28]. Japan was recognized as having a high literacy rate, despite the fact that no literacy survey was conducted in Japan for many years. In addition, the majority of Japanese people can read the labels on food packages/medicine and can understand explanations from medical staff without difficulty. Therefore, scholars considered that the Japanese possess sufficient levels of HL. However, Nakayama et al. concluded, and the current study confirmed, that these levels were inadequate when evaluated using the HLS-EU-Q47. Particularly, they exhibited difficulty in processing health information in terms of appraising the reliability of and applying health information. This result may be partially attributable to the lack of national online platforms that provide neutral, authoritative, comprehensive, and understandable health information. In June 2022, the government announced a policy to establish a Japanese version of the Centers for Disease Control and Prevention of the United States [29]; currently, however, none exists. Therefore, the demand of the Japanese people for appropriate health information has remained unmet. Even without such support, primary care physicians can initially play a role in understanding the healthcare needs of patients and in providing medical information. However, no public scheme exists for people to receive support from physicians or public health nurses as a way to implement primary care from childhood. In addition, the basic school education offered in Japan does not address the HL of students, which may contribute to the inadequate level of HL among the Japanese.

In the present study, being female and in the early stages of university studies were associated with inadequate levels of HL. In previous studies that investigated HL among college students, a few of the researchers concluded that no gender difference exists [30], whereas others found that female students displayed higher levels of HL compared with male students [27,31]. Thus, mixed evidence exists. The inclusion of medical faculties, such as nursing faculties, which have a high proportion of female students as survey targets, may have influenced the results of previous reports in which female students displayed higher levels of HL [31]. On the basis of the results of the quantitative text analysis conducted in the present survey, many female students responded, “Because there is too much information, I’m not sure which one is applicable to me.” Alternatively, the responses of the male students included many descriptions of attitudes toward the use of health information, such as “use multiple sources,” “examine information with a critical eye,” and “always check the source of information.” From these responses, female students appear less confident in judging the applicability of health information for themselves, especially when they obtain a plethora of information. In contrast, male students tend to try to confirm the credibility of the information by exploring a great deal of information. The difference between male and female students in dealing with health information may have contributed to the higher HL levels of male students, compared with those of female students.

One of the reasons that we decided to conduct this survey was concern about students’ perceptions of the COVID-19 vaccine. Few students could explain exactly what “95% vaccine efficacy rate” meant. Notably, some female students believed that they would experience infertility in the future if they received vaccinations for COVID-19. Interestingly, in the present study, the participants who reported receiving the necessary vaccinations exhibited low levels of HL. Previous studies have reported that high levels of HL were associated with high rates of vaccination intention and vaccination [32,33,34]; thus, the results of the current study were contrary to expectations. A recent survey in Brazil used the parental health literacy activities test, the PHLAT-8, to assess the HL of adolescents. The authors reported that the higher the level of HL, the higher the rate of recognition of COVID-19 as a threat. However, the authors found no association between HL level and the willingness to receive vaccinations [35]. The mass immunization of students against COVID-19 at the universities that the participants of the current study attended began in July 2021. Therefore, within the study period, the number of participants who considered the COVID-19 vaccine when responding to the question about vaccination is estimated at <30%. Suzuki et al. recently reported that female students’ willingness to receive a vaccination against human papillomavirus was not associated with HL regarding the HPV vaccine [36]. Their results may stem from the fact that the Japanese government stopped proactively recommending the HPV vaccine from 2013 to 2021. Furthermore, in Japan, people aged < 20 years demonstrated higher rates of vaccination against measles and rubella than those aged > 20 years [37]. Therefore, the government’s campaign to recommend vaccination is considered to have influenced the vaccination rate during the campaign period, which may have led to a high vaccination rate among younger participants with low HL levels in this study. Nevertheless, misinformation regarding the COVID-19 vaccine can discourage people from taking actions that benefit their own health and can unnecessarily inflame anxiety about their daily lives. By providing adequate, easy-to-understand information, the government can significantly help them to make informed decisions regarding vaccination.

As previously mentioned, promoting HL education according to the developmental stage, individual characteristics, and social conditions of the time is important for improving the HL of college students. School education plays a major role, especially during the period leading up to college. In particular, college students undertaking medical courses generally exhibit higher levels of HL than those in nonmedical departments [38,39]. In addition, the more the students undertake health education-related classes, the better their competency in accessing and understanding health information in the health promotion domain [27]. However, providing health education is insufficient for improving HL. According to a systematic review of HL among college students, although the majority responded that they used the Internet to obtain health information, they considered their competencies in searching for health information as *low* and answered that they were unable to find the health information they needed. In particular, they found that critically evaluating information and determining whether or not a data source provided primary or secondary information are difficult aspects to manage. As a result, they displayed negative attitudes toward participation in online health programs and online medical consultations [40]. To address these issues, educational programs that enhance the competencies of college students in examining the information necessary for health promotion, critical thinking about health, and communication skills in relation to health information should be developed. Moreover, practicing such programs that target students in the early grades as standard in educational settings is desirable.

This study has its limitations. First, the HLS-EU-Q47 is a self-reported evaluation tool; thus, the results may deviate from the actual situation in terms of HL. However, the majority of available objective assessment tools consist of questions on competency in reading the information on food labels or calculating accurate doses of medicines. In a case where such objective assessment tools were applied to the current participants, the overall rates of correct responses were expected to be extremely high; thus, individual differences in HL levels cannot be detected. The HLS-EU-Q47 is a widely and internationally used evaluation tool, and the validity of the Japanese version has been verified. Therefore, the study was able to compare HL levels among college students in different countries. We also believe that the results could highlight future challenges in improving HL levels among college students. Second, the HLS-EU-Q47 was conducted via a web survey. Therefore, students who encountered difficulties in responding using smartphones were excluded. This may have influenced the results of the self-assessment of HL regarding Internet use. Furthermore, the majority of the participants were recruited from liberal arts classes at Osaka City University and Osaka Prefectural University. Thus, > 60% were first-year undergraduate students, although they belonged to diverse departments. Generally, first-year college students have just become independent from their parents in terms of living; actually, they show lower levels of HL than those in higher grades of university. This might be the reason why the participants in this study had low levels of HL. Finally, because the survey aimed to evaluate the current state of HL and explore the association between lifestyle or comprehensive health status and HL among Japanese university students, we could not elucidate more details about their sociodemographic data. For example, although the participants were distributed across eight faculties, to what department each student belonged was not considered. However, attributes such as their department, as well as their previous health education, may affect the levels of HL to a non-negligible degree. In the future, clarifying the factors that contribute to HL after adjusting for the effects of such data may provide clues to identify effective intervention targets.

## 5. Conclusions

High levels of HL among college students were associated with more preferable lifestyle habits for health maintenance, along with high levels of subjective health status. The results revealed challenges in facilitating good decision-making for healthcare and promotion throughout life among students with low levels of HL. The challenges need to be kept in mind even as we are gradually returning to normality after the 3-year COVID-19 pandemic. Based on the accumulation of various cases and experiences during the COVID-19 pandemic in Japan and overseas, developing teaching materials for improving the competencies labeled *appraising* and *applying* health information is a necessary aspect. In addition, future studies should determine whether or not such interventions through educational programs can enhance the levels of HL among college students and whether enhancing HL levels can improve health outcomes.

## Figures and Tables

**Figure 1 healthcare-11-00704-f001:**
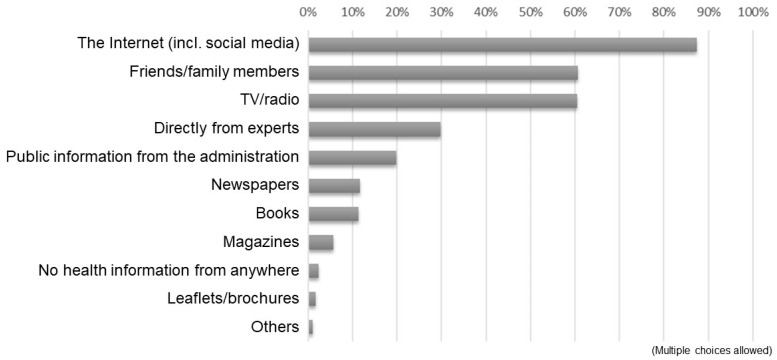
Responses to the question, “Which information source do you use when obtaining health information?”.

**Figure 2 healthcare-11-00704-f002:**
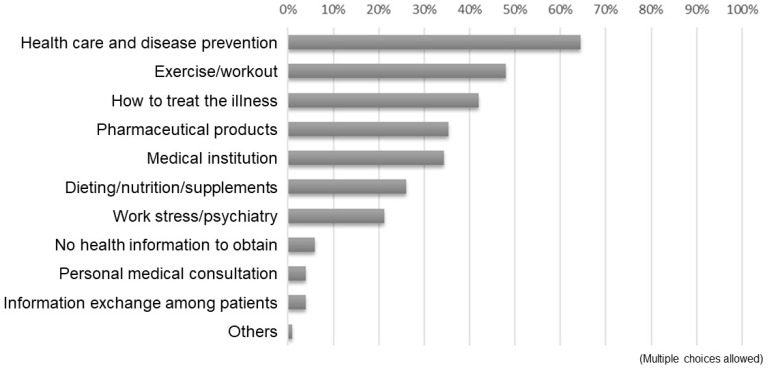
Responses to the question, “What kind of health information do you obtain from the Internet?”.

**Figure 3 healthcare-11-00704-f003:**
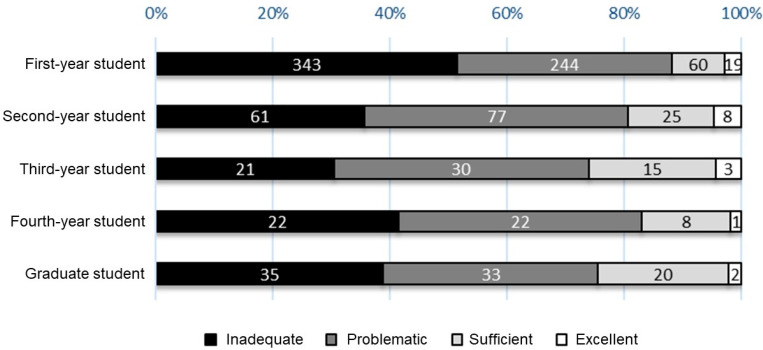
HL levels according to the year level of participating students.

**Figure 4 healthcare-11-00704-f004:**
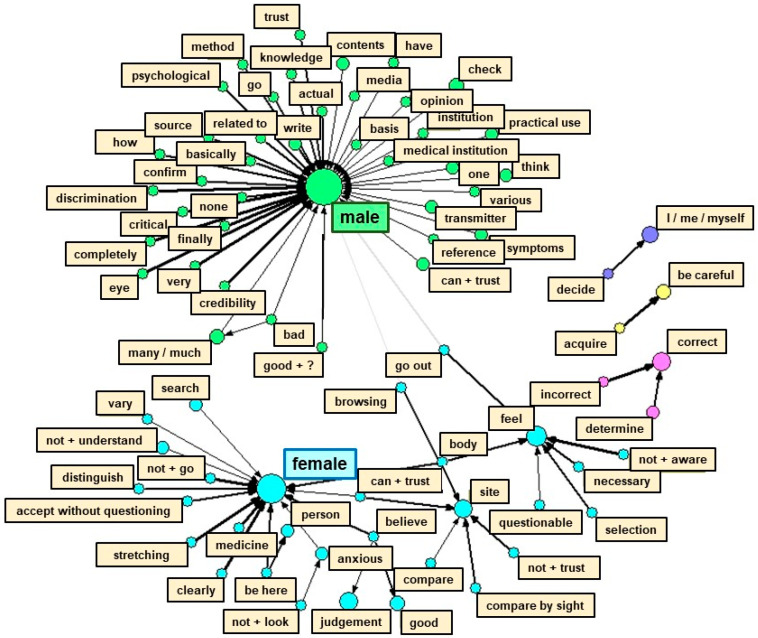
Co-occurrence network schema for visualizing the characteristic terms that frequently appeared in relation to gender in the contents of the open-ended responses. The more frequently the terms appeared together in the sentences, the thicker the arrow; the larger the circle, the more frequently the term appeared.

**Figure 5 healthcare-11-00704-f005:**
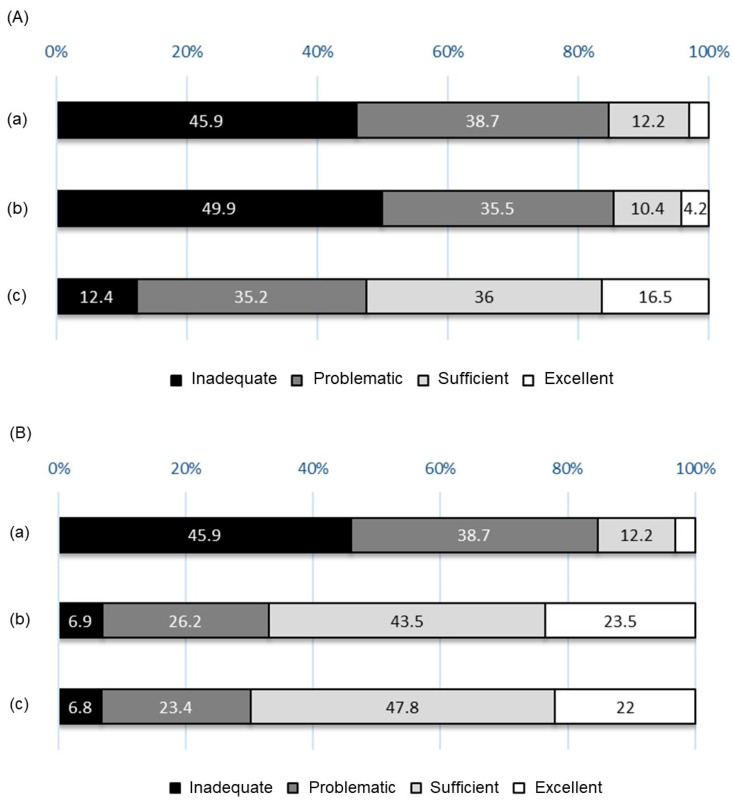
(**A**) Comparison with previous reports on HL levels. (**a**) The results of this study; (**b**) the results from 927 Japanese individuals (age: 20–69 years; [14]); (**c**) the results from 8102 individuals aged 15 years or older, drawn from eight EU countries [14]. (**B**) Comparison with previous reports on HL levels among college students. (**a**) The results of this study; (**b**) the results of 912 students (excluding medical students) from four universities in central Lithuania [27]; (**c**) the results of 381 medical students at the University of Granada, Spain [28].

**Table 1 healthcare-11-00704-t001:** Concepts of the eight health dimensions of SF-36.

Domains	Concepts
Physical functioning	Whether the person can perform various daily activities independently
Role—physical	Whether the person has had problems doing work or daily activities for physical reasons in the past month
Bodily pain	Whether the person has had bodily pain that has prevented him/her from doing his/her usual work in the past month
General health perceptions	Whether the person is in good health and can expect good health in the future
Vitality	Whether the person has been energetic without fatigue over the past month
Social functioning	Whether the person has had physical or psychological issues that interfered with relationships with family, friends, neighbors, or other associates in the past month
Role—emotional	Whether the person has had problems doing work or daily activities for psychological reasons in the past month
Mental health	Whether the person has been calm and happy in the past month

Modified from the manual of the SF-36v2 Japanese version (Fukuhara S and Suzukamo Y. Manual of SF-36v2, Japanese version: iHope International Inc., Kyoto, Japan, 2004, 2019 [19]).

**Table 2 healthcare-11-00704-t002:** Demographic data of the participants.

Total Number	1049
Gender (male/female/preferred not to respond)	623/413/13
Age	19.8 ± 1.9
Year level (first year/second year/third year/fourth year/graduate)	666/171/69/53/90

N or mean ± SD.

**Table 3 healthcare-11-00704-t003:** Mean scores of each competency across the three domains of the HLS-EU-Q47.

	Competencies			
Domains	Accessing	Understanding	Appraising	Applying
Health care	2.6 ± 1.0	2.7 ± 1.1	2.4 ± 1.0	2.8 ± 1.3
Disease prevention	2.7 ± 1.1	3.1 ± 1.8	2.5 ± 0.9	2.7 ± 1.1
Health promotion	2.6 ± 1.0	2.7 ± 1.1	2.5 ± 1.2	2.4 ± 1.1

Mean ± SD. Values are the scores for each competency in the three domains, standardized to a maximum of 5 points.

**Table 4 healthcare-11-00704-t004:** Sentence composition of text data in the open-ended responses.

Item	Value
Total number of responses	368
Mean character count per response	58.1
Total number of sentences	524
Mean character count per sentence	40.8
Total number of terms	4321
Number by term category	1220

**Table 5 healthcare-11-00704-t005:** Top 20 most frequent terms among the contents of the open-ended responses.

Rank	Term	Number of Occurrences
1	judgment	67
2	site	60
2	correct	58
2	I/me/myself	43
3	look	42
3	not + understand	41
3	good	38
3	check	32
3	be careful	31
3	acquire/get	31
3	many/much	29
3	multiple	29
4	write	21
4	can + trust	21
4	confirmation	20
4	disease	18
4	one	17
4	practical use	17
4	credibility	16
4	symptoms	15

**Table 6 healthcare-11-00704-t006:** Content categorization of details in the open-ended responses.

	Category	Examples of Responses Containing Words in Clusters
Common responses of all participants	The difficulty of determining the truth of information	・It is difficult to **distinguish** the truth of medical and health-related information without specialized **knowledge**. ・I **feel **that it is difficult to **distinguish** what is **correct** from a vast amount of information because the information **written** on each **site** is different. ・I **believe** that it is important to **acquire** HL because the Internet is full of **incorrect** information and it is difficult to find **correct** information.
Responses of male students	Looking at information with a critical eye	・Health information obtained from social networking sites and personal blogs is first analyzed with a **critical** eye. ・I am always **critical** of information, no matter what medium it comes from ・I try to be scientific and **critical** about information when the source is not clear.
	Not trusting completely and not believing everything you hear	・I do not **completely trust** the information I receive, but only use it as a reference. ・I don’t think I can **completely trust** the information on the Internet or in the **media**, so I try not to take someone at their word.
	Verifying the source of the information	・I try to confirm the information so that I can identify the **transmitter** of that information. ・I **am careful** to obtain only information from **medical institutions** and government agencies. ・I try to take the time to **check** if the **source** is an **institution **I **can trust**.
	Comparing multiple pieces of information	・I try to **check multiple sites**, not just **one**. ・I **compare multiple sites** to **determine** which statements are facts and which are **opinions**. If **multiple** pieces of information have the same **content**, they are accepted as **correct**.
Responses of female students	Being careful when there are obvious doubts about the credibility of information	I don’t easily **trust** something that is **clearly** different from the health knowledge **I** have gained from school or research papers. I’m trying to **be careful** about **sites** that are **clearly** talking nonsense. I **distrust** advertisements for supplements and other products that are **clearly questionable**.
	Being unable to decide if one can apply the information to oneself	・Even after researching **my symptoms**, I was unable to **determine** if I really had the disease, so it was difficult to take action and actually **go** to a **medical institution**. ・There is so **much** information out there that it’s hard to know what diet will work for **me**. ・I am getting information on **stretching** and weight training, but there is so **much** information that it is difficult to choose what is right for **me**.

Terms with underlined and bold emphases represent those that appeared in the co-occurrence network schema in Figure 4.

## Data Availability

The data presented in this study are openly available on FigShare at doi:10.6084/m9.figshare.21874947.

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
