# Peer review of "Health Literacy among Japanese College Students: Association with Healthy Lifestyle and Subjective Health Status"

_healthcare, 2023, doi:10.3390/healthcare11050704_

Round 1
Reviewer 1 Report
This study reports data from a cross sectional survey of 1049 college students. The study aimed to report the current state of health literacy among university students in Japan, as well as its associations with demographic characteristics and self reported health state. There are several areas that need improvement in the paper:
Methods:
Please report the response rate of the survey? Can you describe how well the sample represents the population (e.g., in terms of gender distribution, subject, year of study)?
Results:
I suggest reorganizing the Results section according to the objectives of the study. At the moment, the results start with information sources that are not mentioned in the study aims proposed in the introduction.
Lines 221-235: Comparison of study results to previous research should occur in the discussion and not in the results. Please consider moving these paragraphs (and probably figures) to the discussion
Lines 239-246: Please describe the findings you present in Fig 5 A-G in some more detail, please refer to individual values in the text as examples. It is useful for the reader, if the authors report the values of HLS-EU-score in the bars in the figures 5 A-G as well as in figures 6 A-H
The data could be used more efficiently. To explore associations of Health Literacy with common health problems, QoL and demographic characteristics, more advanced analysis using methods of inferential statistics (e.g. regression analysis) allowing for adjustment for different confounder can be performed.
Fig 7: Please explain, why you performed gender-specific analysis of open-ended responses, while the rest was not gender-specific
Limitations: Please address the selection bias in the limitations, and whether and to what extent this could influence the survey results.
Author Response
Thank you for reviewing the paper and helpful comments. We would like to address the comments point by point.
Please report the response rate of the survey? Can you describe how well the sample represents L239-247the population (e.g., in terms of gender distribution, subject, year of study)?
According to your suggestion, the descriptions about the response rate and how well the sample represents the population were added as below:.
L187-188
A total of 1,322 students (98.1%) of the 1,347 students invited to participate in the web survey through the classes agreed to participate.
L193-196
The participants were distributed across eight faculties, including humanities and social sciences, science, and medical faculties. The gender distribution also represented the Japanese college student population.
I suggest reorganizing the Results section according to the objectives of the study. At the moment, the results start with information sources that are not mentioned in the study aims proposed in the introduction.
Based on your indication, we added some sentences about the resource of health information to the Introduction section.
L77-84
In 2021, 82.9% of Japanese individuals used the Internet [10]; therefore, promoting the provision of health information via the Internet is necessary. However, because the vast health information available on the Internet is not necessarily reliable, college students need to acquire the skills to judge the health information obtained from the Internet for themselves. Therefore, elucidating the resources for such health information and the current situation of HL among college students would be beneficial for the construction of new schemes for health education that center on HL.
Lines 221-235: Comparison of study results to previous research should occur in the discussion and not in the results. Please consider moving these paragraphs (and probably figures) to the discussion
Your indication is very much appreciated. We moved the paragraph and relevant figures about comparison to previous research to the Discussion section.
Lines 239-246: Please describe the findings you present in Fig 5 A-G in some more detail, please refer to individual values in the text as examples. It is useful for the reader, if the authors report the values of HLS-EU-score in the bars in the figures 5 A-G as well as in figures 6 A-H
Based on your suggestion, we revised the paragraph about the results relevant to the new Fig 4 and 5 as below. We also revised the new Fig 4 and 5 according to your suggestion.
L240-249
Participants who responded that they subjectively had higher general health status showed higher total scores for HLS-EU-Q47 than those who did not (ρ = 0.128, p < 0.001). Similarly, participants who responded that they acquired or practiced healthy lifestyle habits in terms of sleep (ρ = 0.078, p = 0.011), eating (ρ = 0.070, p = 0.024), and maintaining friendships (ρ = 0.064, p = 0.037) had higher total scores for HLS-EU-Q47 than those who did not. Surprisingly, the participants who answered that they received the necessary vaccinations obtained lower total scores for HLS-EU-Q47 than those who did not (ρ = −0.097, p = 0.002). Neither the response to the question about regular physical activity nor smoking habits were correlated to the total scores for HLS-EU-Q47 (Figure 4A–G).
L264-269
The results indicated that high levels of HL were associated with high norm-based scores for physical functioning (ρ = 0.068, p = 0.027), role-physical (ρ = 0.121, p < 0.001), bodily pain (ρ = 0.091, p = 0.003), general health perceptions (ρ = 0.112, p < 0.001), vitality (ρ = 0.104, p < 0.001), role-emotional (ρ = 0.091, p = 0.003), and mental health (ρ = 0.078, p = 0.011). However, HL levels were not associated with norm-based score for social functioning (Figure 5A–H).
The data could be used more efficiently. To explore associations of Health Literacy with common health problems, QoL and demographic characteristics, more advanced analysis using methods of inferential statistics (e.g. regression analysis) allowing for adjustment for different confounder can be performed.
As you pointed out, we could not grasp more details about participants’ sociodemographic data. For example, which department each student belonged to was not taken into account. However, attributes such as the department as well as previous health education could greatly affect the levels of HL. In the future, clarifying the factors that contribute to HL after adjusting for the effects of these confounders using regression analysis may provide clues for identifying effective intervention targets. We added the sentences described above to the limitation paragraph as below:
L449-457
Finally, because the survey aimed to evaluate the current state of HL and explore the association between the lifestyle or comprehensive health status and HL among Japanese university students, we could not grasp more details about their sociodemographic data. For example, although the participants were distributed across eight faculties, what department each student belonged to was not considered. However, attributes such as the department as well as previous health education may affect the levels of HL to a non-negligible degree. In the future, clarifying the factors that contribute to HL after adjusting for the effects of such data may provide clues to identify effective intervention targets.
Fig 7: Please explain, why you performed gender-specific analysis of open-ended responses, while the rest was not gender-specific
Since the difference in the distribution ratio of the levels of HL between male and female was found, we displayed the relationship between gender and terms the co-occurrence network schema for the purpose of exploring the characteristics of responses by gender. We added the sentences described above to the Result section as below:
L292-295
As a difference in the distribution ratio of the levels of HL between male and female was found, we displayed the relationship between gender and the terms to explore the characteristics of the responses by gender.
Limitations: Please address the selection bias in the limitations, and whether and to what extent this could influence the survey results.
As you pointed out, the limitations of this study include selection bias. Students with difficulty in responding using smartphones were excluded. Furthermore, >60% of the participants were first-year undergraduate students who might have not acquired HL compared to those in other grades. This might have been the reason why the current participants had low levels of HL. We added the sentences described above to the limitation paragraph as below:
L440-449
Second, the HLS-EU-Q47 was conducted through a web survey. Therefore, students with difficulty in responding using smartphones were excluded. This may have influenced the results of the self-assessment of HL regarding Internet use. Furthermore, the majority of the participants were recruited from liberal arts classes at Osaka City University and Osaka Prefectural University. Thus, >60% were first-year undergraduate students, although they belonged to diverse departments. Generally. first-year college students have just become independent from their parents in terms of living; possibly, they have not yet acquired HL compared to those in higher grades. Actually, they show lower levels of HL compared to those in higher grades. This might be the reason why the participants had low levels of HL.

Reviewer 2 Report
This is an interesting manuscript on the health literacy of Japanese university students. The comments I would make in this regard would be as follows:
ABSTRACT SECTION
*Please consider entering the abbreviation of HLS-EU-Q47 the first time it appears in the text.
*Please consider including the word "health literacy" as a keyword.
INTRODUCTION SECTION
*Please consider incorporating the abbreviation HL (for health literacy) throughout the document as it appears multiple times.
METHOD SECTION
*Please consider entering the information for the web survey used to collect participant data.
*Although the information provided in the Methods is adequate, I believe that some of this information should be included in the Introduction rather than in this section.
*The information on the characteristics of the HLS-EU-Q47 (lines 110 to 113) should be included in the introduction since this is information that was previously known and obtained by other authors and that the authors of this paper use for their methodology. The same applies to lines 129 to 133.
I recommend that authors consider which parts of the methodology are their own and which information belongs to other authors, and consider including this information in the Introduction or Discussion section. Remember that you can refer to other authors' information in the methodology, if necessary, but you do not need to explain it because the authors have already explained it in their own previously published work.
RESULTS SECTION
* It would be interesting if the authors included a small table with the participants' sociodemographic characteristics since they only present the percentages of prevalent values.
*In the aim of the study stated in the introduction, the authors indicate that they want to examine the factors that influence health literacy. However, despite this and all the results stated, it is not clear which factors will be analyzed and on what basis they chose these and not other factors (e.g., socioeconomic status, previous health education, the field of study of university students, etc.). Authors should make it clear in the methodology section what type of factors they want to analyze in the study and state this in the results.
* Similarly, the authors present part of the discussion in the results section. If the authors compare their own results with those of other authors, this should be included in the Discussion section since it is not part of the objective of the study. Please review the results sections carefully and move any comparisons of your results with those of other authors to the discussion section.
DISCUSSION SECTION
*Please include here the information you have included in your results to compare your results with those of other authors.
*In the limitations, you state that you are unable to determine the relationship between health literacy, subjective health status, and healthy lifestyles because of the characteristics of the study design. If this was not initially possible, you should either adjust the study aim, or you do not need to state that the type of study (cross-sectional) is a limitation, as this may have been known before data collection began.
*Have you considered Social Desirability Bias as a limiting factor?
*Lines 426 to 429 should be included in the results (it would be helpful to have a table of socio-demographic characteristics to know this information).
I encourage authors to implement the comments to make their work more complete and understandable.
Thank you
Author Response
Thank you for reviewing the paper and helpful comments. We would like to address the comments point by point.
*Please consider entering the abbreviation of HLS-EU-Q47 the first time it appears in the text.
We described the abbreviation of HLS-EU-Q47 the first time it appears in the Abstract.
*Please consider including the word "health literacy" as a keyword.
We added "health literacy" as a keyword.
*Please consider incorporating the abbreviation HL (for health literacy) throughout the document as it appears multiple times.
We used the abbreviation HL for health literacy throughout the paper.
*Please consider entering the information for the web survey used to collect participant data.
We added the information about the company we entrusted with the design of the web questionnaire and the collection of the response results to the Method section as follows:
L110-112
We entrusted the design of the web questionnaire and the collection of the responses to Data Select Co., Ltd. (Toyoake City, Aichi, Japan).
*Although the information provided in the Methods is adequate, I believe that some of this information should be included in the Introduction rather than in this section.
*The information on the characteristics of the HLS-EU-Q47 (lines 110 to 113) should be included in the introduction since this is information that was previously known and obtained by other authors and that the authors of this paper use for their methodology. The same applies to lines 129 to 133.
I recommend that authors consider which parts of the methodology are their own and which information belongs to other authors, and consider including this information in the Introduction or Discussion section. Remember that you can refer to other authors' information in the methodology, if necessary, but you do not need to explain it because the authors have already explained it in their own previously published work.
Thank you for your suggestion. As you pointed out, HLS-EU-Q47 is an existing questionnaire and is not our original, but we thought it is necessary to mention the establishment of the Japanese version of the HLS-EU-Q47 and the scoring method we adopted. Therefore, this will be described in the Method section as it was originally. Similarly, we created our own questions about lifestyle habits of college students and use the critical health issues for college students advocated by ACHA only as reference. We think this still needs to be explained in the Method section. We really appreciate your understanding.
* It would be interesting if the authors included a small table with the participants' sociodemographic characteristics since they only present the percentages of prevalent values.
*Lines 426 to 429 should be included in the results (it would be helpful to have a table of socio-demographic characteristics to know this information).
Based on your suggestion, we inserted a new Table 2 about the demographic data of the participants.
*In the aim of the study stated in the introduction, the authors indicate that they want to examine the factors that influence health literacy. However, despite this and all the results stated, it is not clear which factors will be analyzed and on what basis they chose these and not other factors (e.g., socioeconomic status, previous health education, the field of study of university students, etc.). Authors should make it clear in the methodology section what type of factors they want to analyze in the study and state this in the results.
As you pointed out, we thought the descriptions about why the factors were included in the analysis were needed. We added some sentences to the Methods and Results sections as follows:
L134-136
Aiming to investigate the relationship between lifestyle habits that affect health conditions and HL, we investigated the common health challenges faced by college students.
L146-148
Aiming to investigate the relationship between HL and subjective health awareness or comprehensive health status, we investigated the health-related quality of life of the participants.
* Similarly, the authors present part of the discussion in the results section. If the authors compare their own results with those of other authors, this should be included in the Discussion section since it is not part of the objective of the study. Please review the results sections carefully and move any comparisons of your results with those of other authors to the discussion section.
*Please include here the information you have included in your results to compare your results with those of other authors.
Your indication is very much appreciated. We moved the paragraph and relevant figures about comparison to previous research to the Discussion section.
*In the limitations, you state that you are unable to determine the relationship between health literacy, subjective health status, and healthy lifestyles because of the characteristics of the study design. If this was not initially possible, you should either adjust the study aim, or you do not need to state that the type of study (cross-sectional) is a limitation, as this may have been known before data collection began.
According to your comments, we deleted the first part of the limitation paragraph and added the sentences as below to the conclusion instead:
L467-469
In addition, future studies should determine whether or not such interventions through educational programs can enhance the levels of HL among college students and whether enhancing HL levels can improve health outcomes.
*Have you considered Social Desirability Bias as a limiting factor?
In this study, the person in charge of the study could not identify the individual from the responses, and the participants also answered after understanding that. Additionally, participants only had access to aggregated results. Therefore, we did not consider the social desirability bias this time and did not mention it in the limitation paragraph. We appreciate your understanding.

Reviewer 3 Report
Concerning Figures 5 and 6, I recommend in a Note below each figure you to specify which statistical method you used for group comparisons (reported p - values).
This article is interesting and well-written, but there are some minor issues to be clarified further:
Why two p-values are reported for each sub-figure of Figures 5 and 6?
You state "the study used Spearman's correlation analysis to explore the correlation between variables"
I have not found any correlations between the variables reported.
Author Response
Thank you for reviewing the paper and kind comments. We would like to address the comments as follows.
Concerning Figures 5 and 6, I recommend in a Note below each figure you to specify which statistical method you used for group comparisons (reported p - values).
Why two p-values are reported for each sub-figure of Figures 5 and 6?
You state "the study used Spearman's correlation analysis to explore the correlation between variables"
I have not found any correlations between the variables reported.
As we mentioned in the Methods section, this study used Spearman's correlation analysis to explore the correlation between categorical variables and continuous variables (scores). Based on your suggestion, we added the sentence below to the footnotes of the new Fig 4 and Fig 5:
Rho (ρ) indicates the correlation coefficient by Spearman's correlation analysis.
As we described above, the upper value for each sub-figure indicates Rho (ρ), the correlation coefficient, not p.
